# Weil’s Disease in an HIV-Infected Patient: A Case Report and Literature Review

**DOI:** 10.3390/diagnostics13203218

**Published:** 2023-10-16

**Authors:** Xinchun Zheng, Pengyuan He, Ruihua Zhong, Gongqi Chen, Jinyu Xia, Chunna Li

**Affiliations:** Infectious Disease Prevention and Treatment Center, the Fifth Affiliated Hospital of Sun Yat-sen University, Zhuhai 519000, China; zhengxch8@mail.sysu.edu.cn (X.Z.); hepy@mail.sysu.edu.cn (P.H.); zhongrh8@mail.sysu.edu.cn (R.Z.); haizi2308@sina.com (G.C.)

**Keywords:** Weil’s disease, HIV infection, leptospirosis, metagenomic next-generation sequencing

## Abstract

Weil’s disease, an icterohemorrhagic infection, is the most severe and fatal form of leptospirosis and is characterized by jaundice, renal dysfunction, and hemorrhagic predisposition. Weil’s disease with HIV infection has rarely been reported. A 68-year-old male with HIV infection presented to our hospital with fever and dyspnea that progressed to severe hemoptysis and systemic multiple organ failure, necessitating a tracheal intubation ventilator. A diagnosis of Weil’s disease was made after *Leptospira interrogans* was identified via metagenomic next-generation sequencing (mNGS) in bronchoalveolar lavage fluid (BALF). After immediately receiving supportive therapy and targeted antimicrobial agents, the patient achieved complete recovery upon discharge. The co-infection of HIV infection and leptospirosis resulting in systemic multi-organ failure is rare, but awareness should be raised of the differential diagnosis. mNGS can help identify pathogens and facilitate the use of targeted and efficacious antimicrobial therapy in unusual clinical environments.

## 1. Introduction

Leptospirosis is an acute animal-origin infectious disease caused by pathogenic *Leptospira*. Humans become infected through broken skin and mucosal membranes in contact with epidemic water containing *Leptospira* [1]. The clinical manifestations of leptospirosis are extensive but without clear specificity and are difficult to identify; they are characterized as leptospiral sepsis in the early stage, with the damage and dysfunction of various organs occurring in the middle stage, and the various sequela caused by immune-mediated reactions occurring in the final stage. In some cases (approximately 10%), life-threatening manifestations can be presented, such as liver failure, jaundice, hemolysis, acute renal injury, and pulmonary hemorrhage (icteric disease) [2]. Weil’s disease, known as icterohemorrhagic infection, represents the most severe and fatal form of leptospirosis and is characterized by jaundice, a hemorrhagic predisposition, and multi-organ dysfunction including hepatic, renal, pulmonary, cardiac, and central nervous system dysfunctions [3,4]. Leptospirosis in patients with an HIV infection was reported in the West Country [5] and Northern Tanzania [6]; these patients were severely ill compared to rarely reported severity in the East. In China, the earliest cases of leptospirosis date back to the 1930s. *Leptospira interrogans* are thought to be the main cause of leptospirosis [7], although other pathogenic species have also been identified [8,9,10]. The microscopic agglutination test (MAT) is the main standard for diagnosing leptospirosis, as it detects antibodies against *Leptospira*; however, it is very difficult to achieve a timely and accurate diagnosis. Recently, mNGS has been applied to clinical samples and was shown to provide fast and precise pathogen detection and identification [11,12]. In this case, we describe Weil’s disease when complicated by the dysfunction of systemic and multiple organs (including the lung, liver, kidney, heart, pancreas, and central nervous system) in a patient with an HIV infection and summarize the clinical manifestations that have been reported in the literature for patients diagnosed with leptospirosis.

## 2. Metagenomic Next-Generation Sequencing (mNGS)

Sample DNA were obtained via microbial cell wall disruption and fully automatic nucleic acid extraction. A DNA library was constructed via DNA fragmentation, endrepair, adaptor ligation, and polymerase chain reaction (PCR) amplification. High-quality sequencing data were obtained using the BGISEQ-50 platform, and the reads were aligned with Microbial Genome Databases. After receiving the results of taxonomic assignments, reads mapped to *L. interrogans* by MegaBLAST were aligned to the NT database with default parameters for further confirmation. The mNGS was tested by BGI Genomics Co., Ltd. (Shenzhen, China), and was considered positive for *Leptospira* test reads greater than 3.

## 3. Case Report

A 68-year-old male who presented with a cough for 3 days and fever alongside shortness of breath for 1 day visited the emergency department of the Fifth Affiliated Hospital of Sun Yat-sen University on the morning of 6 September 2022. He had a 2-year history of HIV infection with regular treatment using anti-AIDS drugs (3TC + TDF + EFV) and a 20-year history of diabetes with regular oral metformin treatment. Three days before admission, he developed a cough with dark red sputum, mild chest tightness, lower extremity fatigue, bilateral thigh muscle aches, and no nausea or vomiting. One day before admission, he developed a fever (maximum body temperature 39.1 °C) accompanied by shortness of breath and fecal incontinence 3 times. The patient accepted antibacterial and phlegm-reducing therapy after his local hospital outpatient visit. He visited the emergency department of our hospital due to dyspnea exacerbation and was admitted to the emergency intensive care unit (EICU) after an urgent investigation of his chest using computed tomography (CT) showed a high-density shadow and ground-glass opacity alongside exudative inflammatory changes in both lungs (Figure 1D). A physical examination revealed a body temperature of 36 °C, a breathing rate of 28 breaths per minute, a pulse rate of 110 beats per minute, and a blood pressure of 135/82 mmHg. He presented with skin jaundice and icteric sclera (Figure 1A), tachypnea, hemoptysis, bilateral wet rales on the lung’s auscultation, normal heart sounds, and a normal abdominal examination. He was transferred to the infection ICU due to his HIV infection history. Laboratory tests in the morning and afternoon (as shown in Table 1) showed elevated leukocytes, anemia, thrombocytopenia, lymphopenia, elevated procalcitonin (PCT), acute renal failure, hyperbilirubinemia, hypoproteinemia, myocardial injury, acute cardiac insufficiency, coagulation abnormalities, and acute respiratory failure, suggesting the rapid progression and deterioration of the patient’s condition. The patient underwent nasotracheal intubation and ventilator-assisted breathing in the afternoon due to the worsening of dyspnea and low oxygenation index. We then performed a bronchoscopy, which showed a bronchoalveolar hemorrhage in both lungs and a large number of aspirated old blood clots (Figure 1B,C); we collected bronchoalveolar lavage fluid (BALF) to perform mNGS, tested by BGI Genomics Co., Ltd. The ECG showed rapid atrial fibrillation after 2 days (Appendix A). Tests for the tuberculosis antibody, autoantibody, and antinuclear antibodies were negative. The Weil–Felix test for typhoid was negative. Enzyme-linked immunosorbent assay (ELISA) for anti-Leptospira IgG was positive. The CD4^+^ T-cell absolute count was 198/μL, and the CD8^+^ T-cell absolute count was 45/μL. The fungal (1–3)-β-D glucan test was 75.79 pg/mL (normal value < 70). Four days later, BALF mNGS results showed *L. interrogans* and *Aspergillus shevalier* infection (as shown in Table 2).

The therapy process after admission is shown in Figure 2. Based on the history of HIV infection, clinical symptoms, and lung imaging findings, the patient was first suspected to be diagnosed with a bacterial, fungal, or *Pneumocystis jirovecii* infection and was empirically treated with a combined anti-infection regimen of piperacillin/tazobactam (Tazocin), caspofungin, and sulfamethoxazole after admission and accepted methylprednisolone treatment (40 mg once daily) to reduce inflammatory exudation. Three days after admission, Tazocin was switched to meropenem to strengthen anti-infection therapy due to the aggravation of the patient’s condition based on a body temperature increase to 39.8 °C. Five days after hospitalization, penicillin and voriconazole were, respectively, applied to deal with leptospirosis and aspergillus infection when BALF mNGS findings suggested *L. interrogans* and *Aspergillus shevalier* infection. Surprisingly, the patient’s BALF cultured *Enterococcus faecalis* after 11 days; vancomycin was added to treat it, and then a switch was made to linezolid therapy due to elevated creatinine levels. The clinical course of the patient after admission developed into acute respiratory distress syndrome (ARDS), pulmonary hemorrhage, septic shock, acute renal failure, hyperbilirubinemia, and acute cardiac insufficiency; therefore, we provided the patient with mechanical ventilation and supportive treatment, adequate sedation, analgesia and muscle relaxation, norepinephrine to boost blood pressure, hemostasis, bedside continuous hemodialysis (13 times), bedside plasma exchange therapy (2 times).At the same time, an infusion of gamma globulin, human blood albumin, platelets, red blood cells, and fresh human plasma was administered for supportive treatment during hospitalization. As shown in Figure 2, after hospitalization treatment, the patient’s body temperature, white blood cells, neutrophils, platelets, PCT, and CRP levels gradually returned to normal levels (Figure 2A,B); the level of liver function indicators (e.g., ALT, AST, TBil, Dbil) (Figure 2C), cardiac enzymes and cardiac function indicators (e.g., CK, CK-MB, LDH, NT-proBNP) (Appendix A), and coagulation indicators (e.g., PT, APTT, Fib, D-Dimer) (Appendix A) gradually decreased to normal. This slow reduction in creatinine levels and the gradual increase in 24 h urine volume in the patient indicated a gradual improvement in renal function (Appendix A). As the patient’s condition improved, a re-examination of the chest CT after 11 days showed the clear absorption of exudation lesions in the bilateral lungs (Figure 1E), after which the patient was weaned from the ventilator for successful extubation. Unexpectedly, abdominal CT and head CT, respectively, suggested that pancreatic lesions were suspected of harboring pancreatic malignant neoplastic lesions (Figure 1F) and multiple high-density foci in the bilateral cerebral hemispheres; the right parts of the cerebellar hemispheres were suspected of metastasis and bleeding (Figure 1H,L). Interestingly, a repeated abdominal CT examination after 30 days indicated that the pancreatic head mass was smaller than before (Figure 1G), suggesting the possibility of inflammatory lesions and that the level of blood amylase and pancreatic enzymes significantly increased (Appendix A); therefore, we speculated that pancreatic lesions were acute pancreatitis caused by *Leptospira* infection. In the end, the patient was transferred to a general ward on their 23rd day of hospitalization and was successfully discharged after the 40th day of hospitalization. In addition, the patient was followed up one month after discharge, and the head lesion was absorbed according to the head CT re-examination; therefore, we speculated that it was a cerebral hemorrhage caused by *Leptospira* infection.

## 4. Discussion

Leptospiral pathogens are excreted by maintenance hosts such as rodents through their urine, which are maintained in wild and domestic environments due to their transmission among mammalian species (e.g., dogs, cats, cattle, pigs, and goats), presented as a chronic, asymptomatic carriage state [13,14]. When directly in contact with reservoir animals or exposed to surface water or moist soil contaminated with their urine, humans can be infected by *Leptospira* spp. through abrased or damaged skin and mucous membranes. *Leptospires* enter the bloodstream and disseminate throughout body tissues, causing leptospirosis. Leptospirosis typically presents as a biphasic disease and has various protean manifestations. The first phase of this disease is often referred to as the ‘leptospiraemic’ phase, which is characterized by many non-specific clinical symptoms, such as fever, headache, myalgia, conjunctival suffusion, jaundice, cough, hemoptysis, lymphadenopathy, rash, anorexia, nausea, and vomiting. The second phase is the ‘immune’ phase, which involves systemic multiorgan damage and dysfunction with disease progress. The common organs involved are the liver and kidney, and severe leptospirosis, also known as Weil’s disease, is characterized by hepatic leptospirosis; its clinical manifestations include acute hepatitis, hyperbilirubinemia, hypoproteinemia, coagulation abnormalities, and liver failure alongside leptospiral nephropathy [15]. Its clinical manifestations include hypokalemia and acute renal failure characterized histopathologically with acute interstitial nephritis and acute tubular necrosis, with rapid liver and kidney failure and high mortality. It is necessary for rare and unusual manifestations of this illness that clinicians keep in mind relevant epidemiological scenarios, which could result in the involvement of pulmonary, neural, gastrointestinal, cardiovascular, ocular, hematological, skeletomuscular, and other systems [16,17] (summarized in Figure 3). We asked about the medical history of the patient under conscious conditions; the patient was a retiree traveling in Zhuhai city and had a history of wild travel an hiking in grass infested with rodents before hospitalization. Therefore, we suspect that his infection was acquired in wild settings. There are some examples in the literature that document fulminant leptospirosis in patients alongside a co-infection with HIV; these patients were severely ill but responded well to therapy [5,18,19,20]. Similarly, the HIV-infected patient that we report was diagnosed with severe leptospirosis complicated with systemic multiple organ dysfunction, including the lung, liver, heart, pancreas, brain, kidney, and hematological system. It is necessary to raise the alarm that immunocompromised patients may present with more severe clinical manifestations when infected with leptospirosis.

The symptoms of dyspnea and invasive lesions into the bilateral lung using pulmonary imaging on the patient upon admission can easily mislead clinicians to prioritize the diagnosis of AIDS opportunistic infections such as viral pneumonia, invasive Aspergillus pneumonia, and Pneumocystis yehinii pneumonia. After *L. interrogans* and *Aspergillus shevalier* were detected in the patient’s BALF through the mNGS technique, the patient was finally diagnosed with Weil’s disease complicated with pulmonary fungal infection according to their clinical manifestations such as jaundice, acute liver failure, acute kidney failure, and pulmonary hemorrhage. It is a challenge for clinicians to quickly identify pathogen infections in AIDS patients presenting with respiratory symptoms and rapid progression combined with multiple organ failure, especially for some uncommon infectious diseases such as leptospirosis. MAT is considered the reference standard test for the serological diagnosis of leptospirosis, but it does not allow for an early diagnosis due to the dependence and detection of antibodies to the leptospiral antigen and cannot be determined until 5–7 days after exposure. Many studies demonstrate that mNGS is suitable for the detection of pathogenic organisms, especially for rare and slow-growing organisms obtained with difficulty from conventional culture methods [21,47], as mNGS can elucidate organisms and infections at the molecular level. mNGS displays considerable advantages in shortening the time required for the diagnosis of pathogen infections, promoting targeted antimicrobial therapy, and improving patient outcomes [48]. BALF collection is sufficiently tolerated and safe in patients [49]. BALF mNGS improved pathogen detection capability and provided guidance in the clinic which was easy to implement in practice [50]. It is very important that mNGS results should be evaluated in combination with epidemiological and clinical manifestations before pathogenic microorganisms can be identified.

Studies have shown that patients with leptospirosis with severe pulmonary hemorrhagic syndrome have a mortality rate as high as 74% [51], while patients admitted to intensive care units have a 52% risk of death [52]. Therefore, timely diagnosis and timely treatment are the key to reducing the mortality rate for severe leptospirosis. The early clinical suspicion of leptospirosis and laboratory diagnosis is crucial, as delayed diagnosis can increase mortality. mNGS plays an important role in this case and is a powerful and rapid tool for the diagnosis of atypical clinical presentations.

## Figures and Tables

**Figure 1 diagnostics-13-03218-f001:**
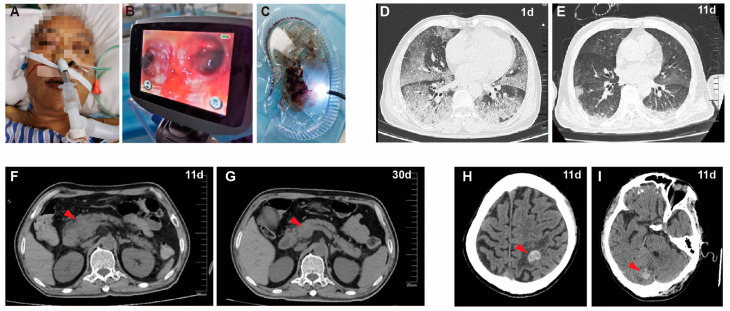
Clinical manifestation and CT imaging of the patient. The imaging shows the skin jaundice of the patient with nasal trachea cannula (**A**), an endobronchial hemorrhage (**B**), and large amounts of dark red blood clots as seen under bronchoscopy (**C**) on 1 day of admission. Chest CT suggested exudative inflammatory changes in both lungs on the first day of admission (**D**) and showed the significantly improved absorption of exudative lesions in both lungs on day 11 (**E**). Abdominal CT showed enlarged pancreatic lesions suspected of malignant pancreatic lesions on day 11 (**F**, red arrow), and a repeat abdominal CT on day 30 showed a smaller pancreas and absorption of pancreatic lesions (**G**, red arrow), suggesting acute pancreatitis. Cranial CT on day 11 showed multiple high-density foci of the brain and cerebellum, suggesting a hemorrhage (**H**,**I**, respectively, red arrow).

**Figure 2 diagnostics-13-03218-f002:**
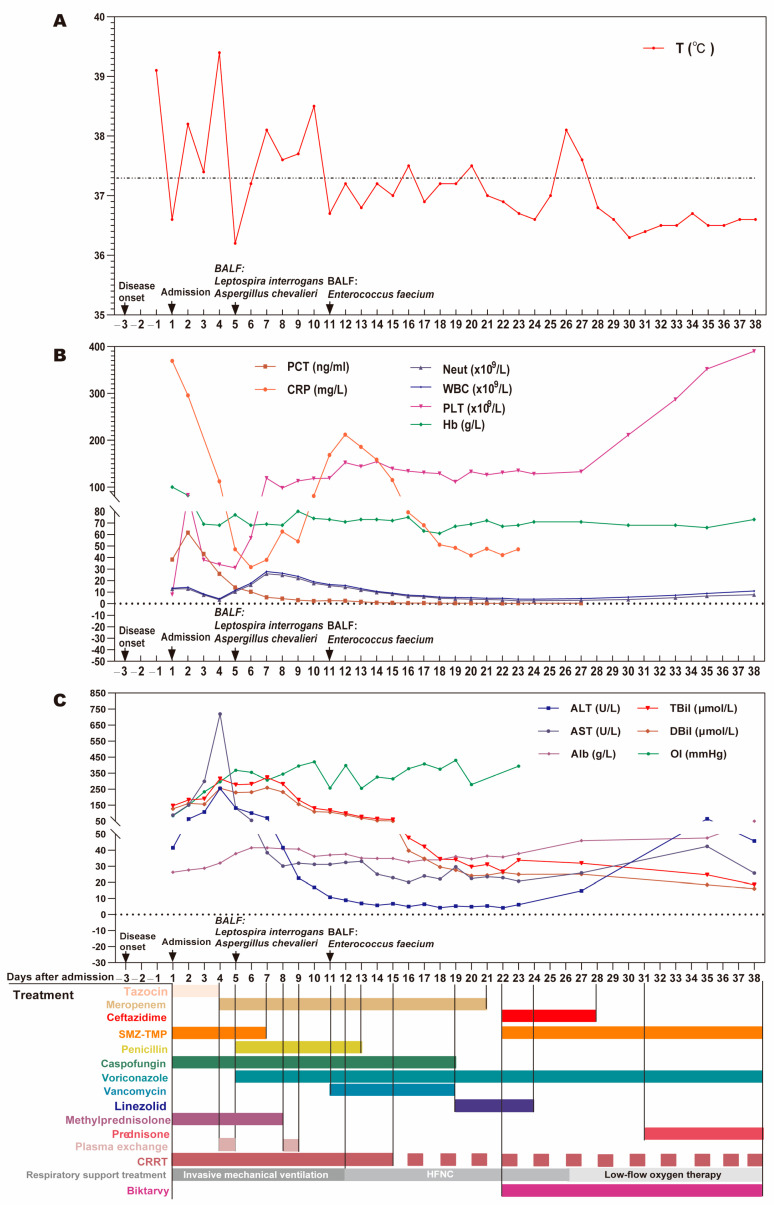
Treatment process and dynamic changes in clinical test parameters of the patient in the clinical course. Dynamic change in the highest daily body temperature (**A**), inflammatory markers (PCT, CRP) and blood routine markers (**B**), liver function indicators (ALT, AST, Alb, TBil, DBil), and oxygenation index (OI) (**C**) during hospitalization.

**Figure 3 diagnostics-13-03218-f003:**
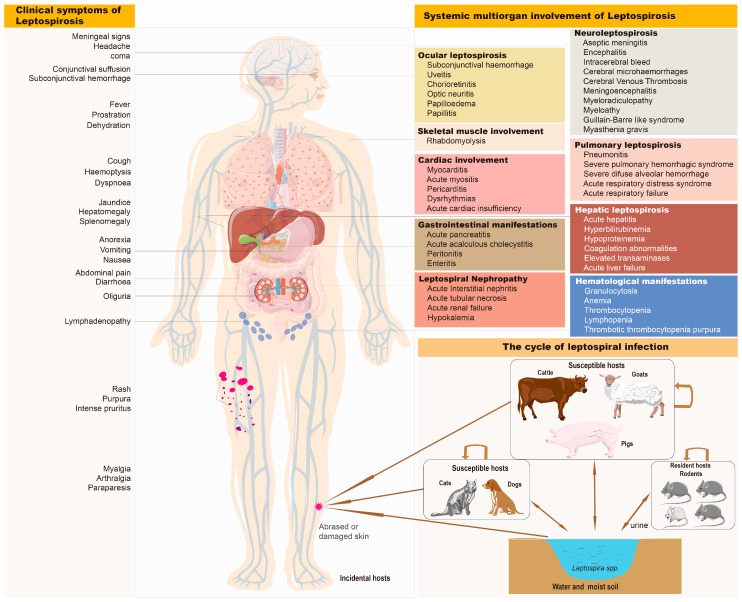
The cycle of leptospiral infection and the clinical manifestations of leptospirosis. Leptospiral pathogens are excreted by maintenance hosts such as rodents through their urine and are maintained in wild and domestic environments due to their transmission among rodent species with a chronic, asymptomatic carriage state. Livestock, domestic, and wild animals can be infected by *Leptospira*, develop a range of disease manifestations and carrier states, and maintain leptospiral infection in these populations due to transmission within animal herds or their continued exposure to rodent reservoirs. When facing direct contact with reservoir animals or exposure to surface water or moist soil contaminated with their urine, humans can be infected by *Leptospira* spp. through abrased or damaged skin and mucous membranes. Leptospira enters the bloodstream and disseminates throughout body tissue, contributing to leptospirosis and showing various clinical symptoms. The invasion of *Leptospira* into various organ tissues causes various clinical manifestations such as neuroleptospirosis [21,22,23,24,25,26,27,28,29], pulmonary leptospirosis [30,31,32,33,34], hepatic leptospirosis, hematological manifestations [35,36,37,38], leptospiral nephropathy [15,39], gastrointestinal manifestations [40,41,42], cardiac involvement [43], rhabdomyolysis [44], ocular and cutaneous leptospirosis [45,46]. Humans are incidental hosts because they do not excrete sufficient numbers of leptospires to serve as reservoirs for transmission.

**Table 1 diagnostics-13-03218-t001:** Laboratory results.

Parameter	2022 September 6 a.m.	2022 September 6 p.m.	Normal Range
White blood cell (WBC) (×10^9^/L)	15.36	13.43	3.5–9.5
Neutrophil (Neut) (×10^9^/L)	14.36	12.68	11.8–6.3
Red blood cell (RBC) (×10^12^/L)	3.19	3.17	3.8–5.1
Hemoglobin (Hb) (g/L)	99	100	115–150
Platelets (PLT) (×10^9^/L)	20	8	125–350
Lymphocyte (Lym) (×10^9^/L)	0.28	0.36	1.1–3.2
Procalcitonin (PCT) (ng/mL)	28.89	38.25	0–0.5
C-reactive protein (CRP) (mg/L)	-	369.23	0.068–8.2
Creatinine (Cr) (μmol/L)	437.6	464.8	41–81
Urea (mmol/L)	20.69	26.15	3.1–8.8
Estimated glomerular filtration rate (eGFR) (mL/min/1.73 m^2^)	11.13	10.35	>90
Alanine aminotransferase (ALT) (U/L)	-	41.4	7–40
Aspartate aminotransferase (AST) (U/L)	92.9	87.6	13–35
Total bilirubin (TBil) (μmol/L)	86.4	146.3	0–21
Direct bilirubin (DBil) (μmol/L)	77.2	127.2	0–8
Albumin (Alb) (g/L)	29.5	26.2	40–55
Creatine kinase isoenzyme-MB (CK-MB) (U/L)	90.6	93.2	0–25
Creatine kinase (CK) (U/L)	2595.7	2947.2	40–200
Lactate dehydrogenase (LDH) (U/L)	592.2	957.6	120–250
Cardiac troponin I (cTnI) (μg/L)	-	0.016	0–0.0229
N-Terminal pro-brain natriuretic peptide (NT-proBNP) (pg/mL)	7080	11,000	0–125
Prothrombin time (PT) (s)	13.6	14.4	9.8–12.1
Activated partial thromboplastin time (APTT) (s)	-	29.6	23.9–31.9
International normalized ratio (INR)	1.19	1.26	0.8–1.15
Fibrinogen (Fib) (g/L)	8.76	6.6	1.8–3.5
D-dimer (μg/mL)	1.54	2.44	0–0.55
FiO_2_ (%)	70	50	-
PaO_2_ (mmHg)	58.1	84.9	83–108
PaCO_2_ (mmHg)	26.7	29	35–45
Oxygenation index (OI) (mmHg)	83.86	121.3	-
pH value	7.325	7.289	7.35–7.45
Lactate (Lac) (mmol/L)	3.8	3.4	0.5–2.22

**Table 2 diagnostics-13-03218-t002:** Pathogenic microorganisms detected using metagenomic next-generation sequencing (mNGS).

Species	Number of Detected Reads
*Leptospira interrogans*	136
*Aspergillus chevalieri*	254

## Data Availability

Raw data are available on request. These data are retained at the Fifth Affiliated Hospital, Sun Yat-sen University, and will not be made openly accessible because of ethical and privacy concerns.

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
