# Peer review of "Weil’s Disease in an HIV-Infected Patient: A Case Report and Literature Review"

_diagnostics, 2023, doi:10.3390/diagnostics13203218_

Round 1

Reviewer 1 Report

This is an interesting case report of an HIV-positive patient who developed a coinfection of Leptospira and Aspergillis. However, the English needs to be improved and a bit more explanation of mNGS results is required.

Line 24 - rodents are the main source of Leptospira but I'm not aware that pigs are a main source - please add a reference for pigs. Certainly pigs are a source but rodents (not just rats) are the main source.

Line 36 - reference #5 is very outdated and many more species have been named. Please update.

Line 55 - please spell out EICU and CT (line 56), PCT (line 63), BALF (line 70), ARDS (line 90)

Line 72 - what does the Weil-Felix test look for?

In line 52 it says that the highest body temp was 39.1 but line 84 gives a temp of 39.8. Please clarify.

Line 133 - what level of reads is considered significant? E. faecalis was grown but not detected by mNGS. Is 134 reads enough to diagnose leptospirosis? Is it enough to determine L. interrogans (species level)? Seems like a very low number of reads. What bioinformatic methods were used to identify L. interrogans?

Not sure if the jaundice photo fully protects the patient's privacy (Figure 1 panel A)

Nice treatment graphic!

Line 199 - MAT is the reference standard (not "gold standard") for SEROLOGICAL diagnosis (molecular diagnosis is best).

Quality of English needs a lot of improvement.

Author Response

We are very glad that you have taken the time to review our articles and have come up with many good questions and given useful suggestions. In view of your questions, we have performed a serious discussion. The following are our answers to your questions. We hope that these point-by-point responses can be able to answer your comments. Please see the attachment.

Thank you very much again for your careful review of our article.

Kind regards,

Xinchun Zheng and Chunna Li

Reviewer 2 Report

1. In introduction, there is no literature on leptospirosis in HIV-1 infection.  It is required to cite several papers on it. For example; Table 1 in Jones S, Kim T: CID 2001:33(1 Sep ),  In particular, please cite whether there is a case in the Orient. 

2. It is recommended to summarize the symptoms or treatment  by the date of hospitalization on the table. 

3. Case report: please divide subheading like Sx, diagnosis and treatment

4. Other minors

line 63: PCT->Procalcitonin (PCT)

line 70:   to performed mNGS-->to perform mNGS

line 73:CD3+CD4+ cells -->CD4+ T cells

          CD3+CD8+ cells-->Cd8+ T cells

line 79: Pneumocystis carinii infection-->Pneumocystis jirovecii (in italic) infection

Table 2: Please delete the column "Genus"

Please describe the name of the company of diagnosed with mNGS.

line 70 to performed mNGS--> to perform mNGS.

line 215:is --> are

Author Response

(The authors gave the same response as above.)

Round 2

Reviewer 1 Report

Improved but still some areas that require attention. Very nice description in the author response of the mNGS reads but it needs to go in the text so the reader can understand. The references are still very outdated and more current references need to be included. There's also a study from Tanzania in 2013 on leptospirosis in HIV patients that would be good to reference. 

Line 24 - leptospirosis here does not need capitalization

Line 28 - I would not define this as an "allergic" reaction - perhaps "immune-mediated" is much better.

Line 38 - "...although other species have been identified" - reference needs to be more current. Ref#7 is outdated and mentions 9 pathogenic species. There are now 40 approved pathogenic species. L. sanjuanensis is the most recently-approved pathogenic species. 

Line 40 - Leptospira should be italicized

LIne 77-78 - please add "for typhoid" after the Weil Felix test and italicize "Leptospira". Also spell out MAT.

Line 78 - is it MAT or ELISA?  MAT detects total antibody, not just IgG. Positive at what level? High titer? By a kit? Low positive could indicate past exposure but was the result high enough to confirm leptospirosis?

Line 79-80 - what does the fungal beta D glucan test of 75 tell us?  Positive for fungus?

Line 146 - "using" is not the correct term here. Excreted by?  Also "resident" should be changed to "maintenance" hosts

Line 160 - Weil's not Wells

Line 170 - Need to describe "wild travel" better. Does this mean sleeping outside/hiking/rafting etc? Was there evidence of rodents or wildlife?

LIne 181 - again, use "maintenance" or "reservoir" host

Line 187 and 190 - italicize "Leptospira"

Much improved but commented on some minor issues, mostly w/ scientific capitalization and italicized names.

Author Response

We are very glad that you have taken the time to review our articles and have come up with many good questions and given useful suggestions again. In view of reviewer's questions, we have performed a serious discussion. The following are our answers to your questions. We hope that these point-by-point responses can be able to answer your comments. Please see the attachment.

Thank you very much again for your careful and professional review of our article.

Kind regards,

Xinchun Zheng and Chunna Li
